# Micropulse Transscleral Cyclophotocoagulation for Glaucoma after Penetrating Keratoplasty

**DOI:** 10.3390/diagnostics12051143

**Published:** 2022-05-05

**Authors:** Mihail Zemba, Otilia-Maria Dumitrescu, Alina-Cristina Stamate, Ileana Ramona Barac, Calin Petru Tataru, Daniel Constantin Branisteanu

**Affiliations:** 1Department of Ophthalmology, “Carol Davila” University of Medicine and Pharmacy, 050474 Bucharest, Romania; mhlzmb@yahoo.com (M.Z.); ramona.barac@live.com (I.R.B.); calinpetrutataru@yahoo.com (C.P.T.); 2Ophthalmology Department, “Dr. Carol Davila” Central Military Emergency University Hospital, 010825 Bucharest, Romania; 3Arena Med Bucharest, 022117 Bucharest, Romania; alina_cristam@yahoo.com; 4Department of Ophthalmology, “Grigore T. Popa” University of Medicine and Pharmacy, 700115 Iasi, Romania; dbranisteanu@yahoo.com

**Keywords:** transscleral cyclophotocoagulation, micropulse, postkeratoplasty, glaucoma

## Abstract

The main objective of the article was to assess the surgical outcome of micropulse transscleral cyclophotocoagulation in patients presenting with glaucoma after penetrating keratoplasty. We conducted a retrospective study that included 26 eyes of 26 patients who presented with glaucoma after penetrating keratoplasty, and who were treated using micropulse transscleral cyclophotocoagulation between January 2017 and December 2020. The surgeries were performed using the Iridex Cyclo G6 MicroPulse P3 Probe. The intraocular pressure, mean number of antiglaucoma medications, visual acuity, corneal status, and postoperative complications were analyzed. The minimum follow-up period was 12 months. The success rate after 12 months was 76.9%. The baseline median intraocular pressure was 29 mm Hg and decreased to 18 mm Hg after 12 months. The median number of antiglaucoma medications was also reduced from three preoperatively to one after one year. In seven cases (29.92%), the visual acuity decreased and, in four cases (15.38%), the corneal graft was not transparent. We concluded that micropulse transscleral cyclophotocoagulation is an effective and safe method for the treatment of glaucoma after penetrating keratoplasty.

## 1. Introduction

Penetrating keratoplasty (PK), especially when performed in complicated cases, is frequently followed by a serious and feared complication—secondary glaucoma. The incidence of glaucoma after PK varies among different studies, mostly because of difficulties in defining it [1,2,3]. In the case of these patients, glaucoma impacts visual function through two distinct mechanisms; firstly, by inducing glaucomatous optic atrophy and, secondly, by reducing the transparency of the corneal graft [4,5].

A gold standard for the treatment of this condition is yet to be found. Medical therapy has frequent contraindications and often becomes inefficient over time [6]. Filtrating procedures, such as trabeculectomy or glaucoma drainage devices, are effective at reducing the intraocular pressure (IOP), but are associated with an increased rate of graft failure, due to intraoperative damage of the corneal endothelium, toxicity of the antimetabolites, and contact between the tube and the endothelium [6,7]. Graft survival seems to be particularly endangered by glaucoma drainage devices as they produce alterations to the haemato–ocular barrier, with subsequent changes in the protein content of the aqueous humor [4].

The use of cyclodestructive procedures, such as cryotherapy and cyclophotocoagulation, has been explored in the treatment of glaucoma post PK, usually after other treatment options have failed, and has been associated with an increased rate of serious complications, especially hypotony and even phthisis bulbi [6]. Micropulse transscleral cyclophotocoagulation (MP-TSCPC) is a form of cyclophotocoagulation in which laser energy is delivered to the ciliary body not in a continuous fashion, but in an intermittent fashion, in cycles consisting of short pulses of energy (the “on” periods), followed by pauses when no energy is delivered (the “off” periods). During “on” periods, the laser energy acts on the ciliary epithelium, while during “off” periods, adjacent tissues are allowed to cool and are thus protected from the alterations resulting from high temperatures. Consequently, MP-TSCPC has a satisfactory efficiency for reducing IOP, while also producing less adverse effects [8,9,10,11].

## 2. Materials and Methods

We conducted a retrospective study that included all patients with PK and secondary glaucoma who were treated using MP-TSCPC between January 2017 and December 2020 in the Ophthalmology Department of the Central Military University Emergency Hospital “Dr. Carol Davila” in Bucharest, and with a minimum follow-up interval of 12 months, which was the primary time point. Follow-up visits were scheduled 1, 3, 6, 9, and 12 months after the intervention. This study is in accordance with the principles of the Declaration of Helsinki (2013) and was approved by the Institutional Review Board of the hospital.

*Patient data.* Patient data can be subdivided into initial and postoperative data. Initial data refer to the preoperative baseline characteristics and consist of demographic data (age and gender), the initial condition for which the PK was performed, the pathogenic mechanism of the secondary glaucoma, previous surgery, crystalline lens status, corneal graft status, the best corrected visual acuity (BCVA), IOP, and the number of antiglaucoma medications used. Postoperative data, recorded at each follow-up visit, were IOP, BCVA, number of antiglaucoma medications, corneal graft status, the presence and nature of complications, and the necessity for reintervention. IOP was measured using Goldmann applanation tonometry (Haag-Streit AG, Koeniz, Switzerland) or an iCare rebound tonometer (Tiolat Oy, Helsinki, Finland) when applanation tonometry could not be performed.

*Treatment.* The surgical procedures were performed in the operating room, under retrobulbar anesthesia with a mixture of 3 mL of lidocaine 4% and 1 mL of bupivacaine 1%. We used 2% methylcellulose as a coupling agent to facilitate the slow and continuous movement of the cyclophotocoagulation probe and to enhance the transmission of laser energy to the concerned tissues. MP-TSCPC was performed using a MicroPulse P3 handpiece with the Iridex Cyclo G6 (IRIDEX, Mountain View, CA, USA). The parameters used were a power of 2000 mW and a duty cycle of 31.35% (an “on” period of 0.5 ms and an “off” period of 1.1 ms). The probe was applied on the sclera for a period of 90 s per hemiglobe, using a moderate but firm pressure, in a continuous sweeping motion, and avoiding the 3 and 9 o’clock meridians; filtering blebs; areas of scleral thinning; and glaucoma drainage devices. All surgeries were performed by the same surgeon (M.Z.). Postoperative treatment included topical dexamethasone 0.1%, 4 times daily for 4 weeks, and cyclopentolate 1%, 2 times daily for 2 weeks, as well as preoperative topical antiglaucoma medication (but discontinuing oral acetazolamide). Therapy was adjusted at each follow-up visit.

*Follow-up.* Patients were examined the next day after the intervention and then after 1, 3, 6, 9, and 12 months. After one year, follow-up visits were scheduled at 3 month intervals. For the inclusion of patients in the study, a follow-up of a minimum of 12 months was required.

*Outcome measures.* The primary outcome measure was the successful decrease in IOP. Therapeutic success at any time point was defined as an IOP greater than 5 mm Hg and lower than 21 mm Hg, or a reduction in IOP of more than 30% compared with the baseline value. Hypotonia, defined as an IOP lower than 5 mm Hg, was considered a therapeutic failure. Secondary outcome measures were the number of glaucoma medications, the use of oral acetazolamide, the BCVA, the corneal graft status, and the occurrence of complications.

*Statistical analysis.* All the data from the study were analyzed using IBM SPSS Statistics 25 and were illustrated using Microsoft Office Excel/Word 2013. Quantitative variables were tested for normal distribution using the Shapiro–Wilk test and were written as averages with standard deviations or medians with interquartile ranges (IQRs). Qualitative variables were written as counts or percentages.

Quantitative variables with a non-parametric distribution were tested between intervals using Wilcoxon/Friedman tests. Qualitative variables were tested between intervals using Cochran’s Q test. Post hoc Dunn tests with Bonferroni correction were done to further detail the results obtained in the initial tests. A significance level of 0.05 was selected as the threshold for statistical significance.

## 3. Results

During the analyzed period, 29 patients with post PK glaucoma were treated using MP-TSCPC. Two patients were excluded because of a short length of follow-up (one and three months), while one patient suffered a perforating eye injury requiring evisceration. Finally, 26 eyes of 26 patients were included in the study.

### 3.1. Initial Characteristics

The median age was 67 years (IQR: 54.71–71.25 years). The gender distribution was 15 men (57.7%) and 11 women (42.3%). The main indication for PK was bullous keratopathy in 16 (61.53%) cases. The most frequent causes of glaucoma were the presence of peripheral anterior synechia, steroid treatment, and decompensated preexisting glaucoma. Twenty-four (92.3%) patients had all corneal sutures removed before the MP-TSCPC procedure, while the remaining two still had the running suture. Most of the included patients were complex cases, requiring multiple interventions in addition to PK, including cataract extraction (the most frequent), trabeculectomy, anterior vitrectomy, posterior vitrectomy, wound suture, and iridectomy. At the beginning of the study, in eight patients (30.7%), the corneal graft was not transparent. The baseline patient characteristics are presented in Table 1.

### 3.2. Follow-Up

The median value for the follow-up period was 13.5 months (IQR: 12–15 months, range: 12–24 months).

### 3.3. Primary Outcome—IOP Evolution

Data from Table 2 and Figure 1 show the descending evolution of intraocular pressure over the follow-up period. From a baseline median value of 29 mmHg (IQR: 26–32), the IOP values significantly decreased to a median value of 19 mmHg (IQR: 16–21.25) after 1 month (*p* < 0.001), and remained lowered after 3 months (median = 17 mmHg, IQR: 15.75–19.25) (*p* < 0.001), 6 months (median = 16.5 mmHg, IQR: 13.5–20.5) (*p* < 0.001), 9 months (median = 16 mmHg, IQR: 13.75–18.25) (*p* < 0.001), and 12 months (median = 18 mmHg, IQR: 16–20.25) (*p* = 0.001), according to the Friedman’s test and Dunn–Bonferroni post hoc tests. There was a significant decrease in IOP values from baseline to all of the registered intervals, while the differences between intervals were not statistically significant (*p* > 0.05), showing a stationary and maintained tendency for the evolution of IOP values after surgery. The success rate of IOP reduction after 12 months was 76.9%. The median difference of IOP values from baseline to 12 months was 10 mmHg (IQR: 7–12 mmHg; *p* = 0.001).

### 3.4. Secondary Outcomes

#### 3.4.1. Antiglaucoma Medication

In terms of medication usage, the evolution was similar, according to the data from Table 2 and Figure 2. From the baseline median value of three medications (IQR: 2–3), the number of medications used did not change significantly after 1 month (median = 2 medications; IQR: 0.75–3) (*p* = 0.128), but a significant decrease started from 3 months (median = 1.5 medications, IQR: 0–2) (*p* < 0.001), with the values remaining lowered after 6 months (median = 1.5 medications, IQR: 0–3) (*p* = 0.001), 9 months (median = 2 medications, IQR: 0–3) (*p* = 0.006), and 12 months (median = 1 medication, IQR: 0.75–3) (*p* = 0.013), according to the Friedman’s test and Dunn–Bonferroni post hoc tests.

There was a significant decrease in terms of medication usage starting from 3 months, while the differences between intervals were not statistically significant (*p* > 0.05), showing a stationary and maintained tendency for the evolution of medication usage after surgery. The median difference of medications used from baseline to 12 months was one medication (IQR: 0–2; *p* = 0.013).

The use of oral acetazolamide was lowered even more over time, according to the data from Table 2 and Figure 3. At baseline, 46.2% of patients used acetazolamide; this frequency was significantly lower after 1 month (7.7%; *p* < 0.001), and remained lowered after 3 months (3.8%; *p* < 0.001), 6 months (11.5%; *p* = 0.001), 9 months (15.4%; *p* = 0.007), and 12 months (11.5%; *p* = 0.001), according to the Cochran’s Q test and Dunn–Bonferroni post hoc tests. There was a significant decrease in acetazolamide usage from baseline to all of the registered intervals, while the differences between intervals were not statistically significant (*p* > 0.05), showing a stationary and maintained tendency for the evolution of acetazolamide usage after surgery. From the 12 patients who initially used acetazolamide, at the 12-month mark, only three patients were using acetazolamide, indicating a reduction of 75% in terms of oral acetazolamide usage.

#### 3.4.2. Visual Acuity

In the majority of cases (18; representing 69.23%), BCVA remained unchanged. In seven (26.92%) cases, BCVA decreased during follow-up and, in one case (3.84%), it was improved. The results are presented in Table 3.

#### 3.4.3. Corneal Graft Status

At baseline, 18 corneal grafts were clear and 8 were opaque. After 12 months, 21 patients (80.7%) had an unchanged status of their corneal transplant; in four cases (15.38%), there was opacification of an initially clear graft and, in one case (3.84%), an initially edematous graft became clear.

#### 3.4.4. Complications

Postoperative complications, mostly mild and self-limiting and occurring in the early postoperative period, included subconjunctival hemorrhage (12 cases), mild anterior segment inflammation (3 cases), transitory increase in IOP (2 cases), and corneal abrasion (2 cases). There was only one case of ocular hypotony, occurring in a patient who had two treatment sessions of MP-TSCPC, with an initially incomplete response after the first session and hypotony after the second treatment session.

Reintervention was necessary in five cases—in two cases after 6 months and in three cases after 9 months from the first session of MP-TSCPC. After the second session, in two cases, therapeutic success was obtained; in two cases, the response was unsatisfactory; and in one case, hypotony occurred.

## 4. Discussion

Glaucoma appearing in eyes that have undergone PK is a diagnostic and therapeutic challenge. IOP is difficult to measure; BCVA is often poor, which makes it difficult to evaluate the visual field; and sometimes ocular media are not transparent. The mechanisms that lead to glaucoma are complex as multiple factors are involved. Initially, the increase in IOP can be due to angle distortion, pupillary block, pigmentary dispersion, and hyphema [3,12]. In the long term, glaucoma may occur as a result of chronic angle closure through PAS, prolonged steroid treatment, and chronic inflammation [3,13]. Multiple surgeries of the anterior segment may produce anatomical changes that interfere with aqueous humor outflow.

The patients included in our study had had multiple surgeries, with a mean of 1.53 surgeries per eye. Only four patients were naïve to ocular interventions. What is more, for most patients, the pathogeny of glaucoma included multiple mechanisms (see Table 1).

In our study, MP-TSCPC had a success rate of reducing the IOP by 76.9% after 12 months. We defined success as an IOP between 5 and 21 mm Hg or a reduction in IOP of more than 30% from baseline, regardless of the need for glaucoma medication. An IOP lower than 5 mm Hg was considered a therapeutic failure. Owing to the different definitions of success across different studies that evaluate antiglaucoma interventions, comparing the success rates of different studies is problematic. Some studies define success as a decrease in IOP of more than 20% from the baseline value, while others define it as more than 30%. In this study, we opted for the latter, as we considered that a more significant decrease in IOP was indicated in the case of these patients. This is because, after penetrating keratoplasty, it is difficult to assess the severity and progression of glaucoma using conventional methods (clinical examination of the optic nerve and of the nerve fiber layers, automated perimetry, and optical coherence tomography) because of the partial or total opacification of the corneal graft and a low visual acuity. In these situations, IOP is one of the few objective parameters that can be used for follow-up, although even IOP measurement is subject to many sources of error [14]. Our results are comparable to those of other studies that evaluate the IOP-lowering efficiency of MP-TSCPC, both in patients post PK and in other clinical settings. Tan et al., who had a similar definition of success as in our study, reported a success rate of 80% after 18 months, in a study that included 40 patients with glaucoma of diverse etiologies [15]. Williams and colleagues had a success rate of 67%, with success being defined as an IOP between 6 and 21 mmHg or a decrease in IOP of more than 20% from baseline [10]. Aquino et al. reported a success rate of 73.3% after 12 months [16], Zaarour et al. reported a success rate of 72% after 12 months [17], and Preda et al. reported a success rate of 65.63% after 18 months [18]. In a study by Subramanian et al., which included patients that had undergone PK, the success rate, defined as an IOP lower than 21 mm Hg or a decrease in IOP of more than 20% from baseline, was 72% [4], which was similar to that of Lee and colleagues (70%), who had a similar patient selection and definition of success [6]. We believe that the relatively high success rate in our study was due to MP-TSCPC being used early in the course of the disease, at relatively low IOPs, as 18 patients (69.23%) had IOP values under 30 mm Hg. Consequently, a final IOP of less than 21 mm Hg was more easily achieved.

MP-TSCPC reduced the number of antiglaucoma medications required for IOP control. From an initial median value of 3, the number decreased to a median of 1 after 12 months. Discontinuing oral acetazolamide therapy has great clinical importance and our results were even more satisfying with regard to this aspect. From an initial 12 patients who were dependent on acetazolamide for controlling their IOP, the number of patients dropped to only one after 3 months, with a final number of three patients after 12 months. Our results are similar to those of other studies, such as those by Aquino et al. [16] and Zaarour et al. [17]. Ocakoglu and colleagues reported a reduction in the number of antiglaucoma medications used from a mean of 2.8 to a mean of 1.2 after 12 months [19].

Regarding visual acuity, the patients included in our study had a poor baseline BCVA. In 18 cases, BCVA remained unchanged; in 7 cases (26.92%), it worsened; and in 1 case, it improved. A decrease in BCVA in the case of these patients may be as a result of multiple factors, namely, loss of graft transparency; evolution of the glaucomatous optic neuropathy; or complications related to MP-TSCPC, such as inflammation, corneal erosions, or choroidal effusions. In our study, we considered that MP-TSCPC had a contribution to vision worsening in two cases, one in which there was postoperative hypotony and one in which postoperative inflammation may have resulted in corneal graft failure. Lee and colleagues, in their study that included 30 patients with glaucoma post PK, did not report any cases with a decrease in BCVA after MP-TSCPC [6]—results similar to those of Elhefnev et al. [20]. Aquino et al. reported a 4% rate of BCVA worsening [16].

In our study, we had four cases (15.3%) in which an initially clear graft became opaque and one case in which an initially edematous graft regained clarity as a result of lowering of the IOP. In only one of the four cases was MP-TSCPC believed to have possibly contributed to graft failure as a result of postoperative inflammation. In all other cases, the underlying cause was the low endothelial cell count of the transplanted cornea. In one case, there was also an Ahmed valve tube present, which may have precipitated corneal decompensation. Lee et al. reported a single case of corneal opacification among their 30 patients with glaucoma post PK, but this was due to graft rejection [6]. Subramanian et al., in a study that included 61 eyes that had undergone PK, showed that 94% of grafts retained transparency after one year [4]. Older studies had a higher rate of corneal transplant opacification, between 11 and 44%, but the cyclophotocoagulation technique used was continuous wave and not micropulse [21,22,23].

Other surgical methods used in the management of glaucoma in the setting of PK include trabeculectomy and glaucoma drainage devices. Trabeculectomy, especially when associated with antimetabolites, is effective at lowering IOP. However, it increases the risk of graft failure, either by intraoperative injury to the corneal endothelium or by antimetabolite toxicity, which may lead to corneal ulcers and opacification [6,7]. Glaucoma drainage devices, such as the Ahmed glaucoma valve, the Molteno and the Baerveldt devices, have long been used in patients with glaucoma after PK. They efficiently reduce the IOP, with reported success rates of 71% after 5 years [24] and 70% after 10 years [25]. They appear to be more effective than trabeculectomy at lowering IOP in post PK patients, but are associated with a greater loss of corneal endothelial cells [26]. Graft survival is endangered for several reasons. First, glaucoma drainage devices lead to alterations in the blood–aqueous barrier, which lead to changes in the protein content of the aqueous humor [27]. Second, tube positioning in the anterior chamber may result in contact with the corneal endothelium, with subsequent damage and endothelial cell loss [28]. The percentage of clear corneal grafts varies between 26 and 55% [24,25]. The Ex-PRESS mini-shunt is also an option, and has proven to be non-inferior to trabeculectomy in terms of IOP-lowering capacity [27]. In a study by Ates et al., the postoperative IOP was successfully reduced by 30% or more compared with the baseline values, and no negative effects on the corneal graft were noted [29]. Additional studies comparing these methods and MP-TSCPC would be useful for helping surgeons decide which options are better suited for their particular patients. Moreover, future prospective studies of MP-TSCPC in post PK patients, with the inclusion of a greater number of patients, with standardized criteria for patient evaluation, and with longer follow-up periods, are needed in order to provide results with superior statistic relevance.

A limitation of our study is the small number of cases, which makes it difficult for statistical significance to be attained. This is the result of the scarcity of PK being performed in Romania, resulting in a smaller number of cases of glaucoma appearing in this setting. Another obvious limitation is the retrospective nature of the study. However, collecting the relevant data was possible, as all patients were treated at the same hospital, by the same surgeon, and following a standard protocol of care. Another limitation of the study is that it did not analyze the effect of the procedure on the number of corneal endothelial cells, but only the clinical appearance of the graft.

## 5. Conclusions

MP-TSCPC is a treatment method that has been proven to be effective in cases of glaucoma occurring in the setting of previous PK. The procedure has a good safety profile, minimally affecting visual acuity and the corneal graft.

## Figures and Tables

**Figure 1 diagnostics-12-01143-f001:**
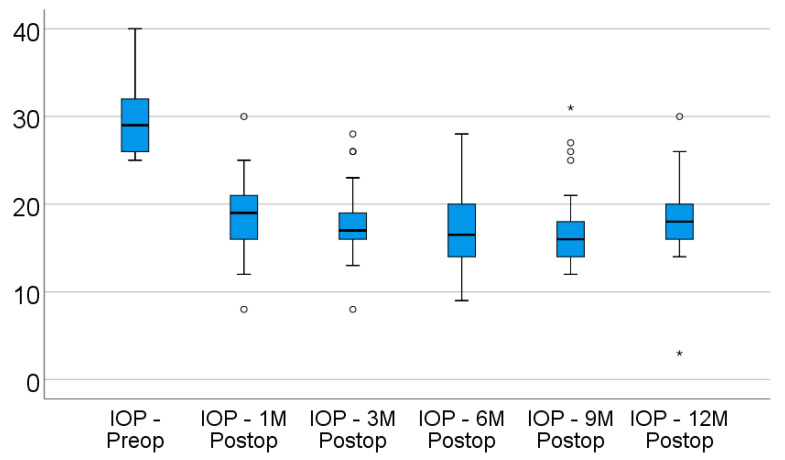
Box-plot representation of the evolution of the intraocular pressure. “*”, “o” represent extreme values (outliers): “*” represents values that are smaller than Q1 (the first percentile) minus 3 times the IQR or greater than Q3 (the third percentile) plus 3 times the IQR; “o” represents values that are smaller than Q1 (the first percentile) minus 1.5 times the IQR or greater than Q3 (the third percentile) plus 3 times the IQR.

**Figure 2 diagnostics-12-01143-f002:**
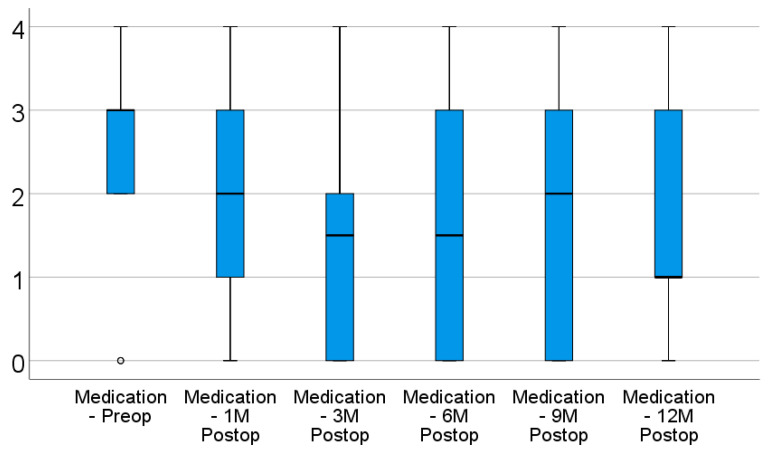
Box-plot representation for the evolution of medication usage. “o” represents an extreme value (outlier) that is smaller than Q1 (the first percentile) minus 1.5 times the IQR or greater than Q3 (the third percentile) plus 3 times the IQR.

**Figure 3 diagnostics-12-01143-f003:**
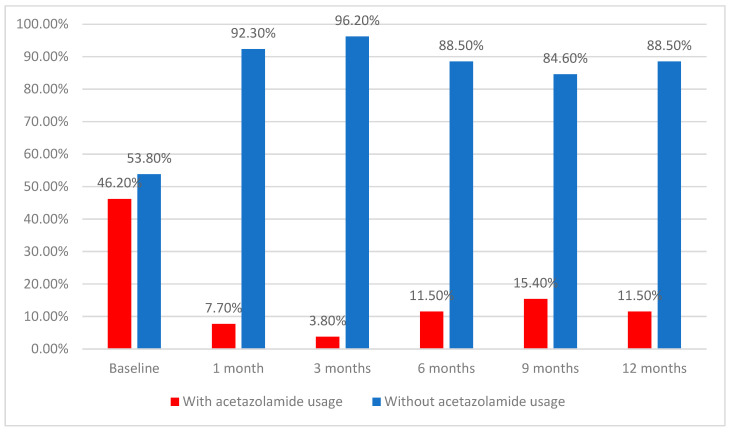
Evolution of acetazolamide usage.

**Table 1 diagnostics-12-01143-t001:** Baseline patient characteristics.

Patient	Gender	Age (Years)	Reason for PK	Cause of IOP Rise	Previous Surgery	Lens Status	Graft Status (Clear)	Follow-Up Period (Months)
1	M	72	BK	PreG, PAS	CE, Trab	PC-IOL	No	12
2	M	68	BK	PreG, ST	CE	PC-IOL	Yes	12
3	F	65	FCD	PreG, ST	CE, Trab	PC-IOL	No	15
4	M	49	PTL	PAS	WC, CE	SF-IOL	Yes	15
5	F	69	BK	PB, PAS	CE, AV	SF-IOL	Yes	12
6	F	71	BK	PB, PAS	CE, AV	SF-IOL	Yes	15
7	M	80	BK	PAS	CE, Trab	PC-IOL	Yes	18
8	F	73	BK	PreG, ST	CE	PC-IOL	No	12
9	M	25	BK	ST	CE	PC-IOL	Yes	12
10	F	64	BK	PAS, ST	CE, Trab, AV	PC-IOL	Yes	15
11	M	59	BCU	PAS; Chr.Inf.	CE	PC-IOL	No	18
12	M	55	GCD	ST	None	Phakic	Yes	24
13	F	78	FCD	PreG, ST	None	Phakic	Yes	12
14	M	31	HK	Chr.Infl.	None	Phakic	Yes	12
15	F	66	BK	ST	CE, Trab-M	PC-IOL	No	24
16	F	75	BCU	PAS, CI	CE	PC-IOL	Yes	12
17	M	70	BK	PreG, ST	CE	PC-IOL	Yes	15
18	F	29	BK	ST, Aphakia	CE	Aphakic	No	15
19	M	44	HK	Chr.Infl., PAS	None	Phakic	Yes	12
20	F	77	BK	ST	CE	PC-IOL	Yes	12
21	M	71	BK	PreG, ST, PAS	PV, CE, SO, AV	PC-IOL	Yes	15
22	M	69	PTL	PAS	WC, CE, PI	PC-IOL	Yes	18
23	M	63	BK	PAS, RI	PV, CE	PC-IOL	No	15
24	M	54	PTL	PAS, ST	WC, Trab-M	Phakic	Yes	12
25	F	70	BK	PreG, ST	CE	PC-IOL	Yes	12
26	M	62	BK	PAS, RI	PV, SO, CE	PC-IOL	No	12

AV = anterior vitrectomy, BK = bullous keratopathy, BCU = bacterial corneal ulcer, CE = cataract extraction, Chr.Infl. = chronic inflammation, F = female, FCD = Fuchs corneal dystrophy, GCD = granular corneal dystrophy, HK = herpetic keratitis, M = male, PAS = peripheral anterior synechia, PB = pupillary block, PC-IOL = posterior chamber intraocular lens, PI = peripheral iridectomy, PreG = preexisting glaucoma; PTL = posttraumatic leukoma, PV = posterior vitrectomy, RI = rubeosis iridis, SF-IOL = scleral fixated intraocular lens, SO = silicone oil tamponade, ST = steroid treatment, Trab = trabeculectomy, Trab-M = trabeculectomy with Mitomycin C, WC = wound closure.

**Table 2 diagnostics-12-01143-t002:** Evolution of IOP, number of antiglaucoma medications, and acetazolamide usage.

Parameter/Time	IOP (Median (IQR))	Medication (Median (IQR))	Acetazolamide Usage (Nr., %)
Baseline	29 (26–32)	3 (2–3)	12 (46.2%)
1 month	19 (16–21.25)	2 (0.75–3)	2 (7.7%)
3 months	17 (15.75–19.25)	1.5 (0–2)	1 (3.8%)
6 months	16.5 (13.5–20.5)	1.5 (0–3)	3 (11.5%)
9 months	16 (13.75–18.25)	2 (0–3)	4 (15.4%)
12 months	18 (16–20.25)	1 (0.75–3)	3 (11.5%)
*p*	<0.001 *	<0.001 *	<0.001 **

* Related samples’ Friedman’s two-way analysis of ranks, ** related samples’ Cochran’s Q test.

**Table 3 diagnostics-12-01143-t003:** Evolution of BCVA and the corneal status.

Patient	Preoperative BCVA	BCVA after 12 Months	Clear Graft after Baseline	Clear Graft after 12 Months
1	HM	HM	No	No
2	6/60	6/10	Yes	Yes
3	CF	HM	No	No
4	6/20	6/20	Yes	Yes
5	6/60	6/10	Yes	Yes
6	CF	HM	Yes	No
7	6/30	6/30	Yes	Yes
8	CF	6/120	No	Yes
9	6/30	6/30	Yes	Yes
10	CF	CF	Yes	Yes
11	CF	HM	No	No
12	6/15	6/15	Yes	Yes
13	6/20	6/20	Yes	Yes
14	6/120	CF	Yes	No
15	CF	CF	No	No
16	6/30	6/30	Yes	Yes
17	6/60	6/60	Yes	Yes
18	HM	HM	No	No
19	6/20	6/20	Yes	Yes
20	CF	HM	Yes	No
21	6/120	CF	Yes	No
22	6/20	6/20	Yes	Yes
23	HM	LP	No	No
24	6/30	6/30	Yes	Yes
25	6/60	6/60	Yes	Yes
26	HM	LP	No	No

CF = counting fingers, HM = hand motion perception, LP = light perception.

## Data Availability

Not applicable.

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
