# Peer review of "Micropulse Transscleral Cyclophotocoagulation for Glaucoma after Penetrating Keratoplasty"

_diagnostics, 2022, doi:10.3390/diagnostics12051143_

Round 1

Reviewer 1 Report

The paper entitled “Micropulse Transscleral Cyclophotocoagulation for Glaucoma 2 after Penetrating Keratoplasty” is a retrospective study that assesses the surgical outcomes of micropulse transscleral cyclophotocoagulation in patients presenting with glaucoma after penetrating keratoplasty. Alternative methods for managing IOP elevations in difficult patients can prove to be of clinical use and help in the management of patients with glaucoma, which remains to be an important sight-threatening disease, especially in patients that have undergone other types of surgery.

The retrospective study has been correctly planned. It is interesting and of clinical interest. The study provides therapeutic options for patients at risk with IOP increases, which adds to current literature in the field of glaucoma.

There are, however, a few minor points that need to be addressed:

  1. Lines 96-100: the authors should explain definitions and reasons for primary outcome measure criteria, with appropriate references to support these choices.
  2. Mention should be made regarding the type of glaucoma and severity, preferably using visual field results and or other widely used glaucoma staging systems to better comprehend the cohort considered.
  3. Details regarding prospective future studies in this field should be included in the Discussion section considering the small group of patients followed for a short time period and the retrospective nature of the study.

Author Response

Answer to Reviewer

We would like to thank the reviewer for taking the time to read and asses our manuscript. We believe the suggestions have helped us improve our study. We have addressed all the concerns that were raised. Modifications are written in red in the new version of the manuscript and we used the Track Changes function of the Microsoft Word. We hope that the revised version meets your requirements. 

  1. Comment: “Lines 96-100: the authors should explain definitions and reasons for primary outcome measure criteria, with appropriate references to support these choices.”

Response:

Our definition of success is similar to that of other studies, mentioned in the “Discussion” section. The 5-21 mm Hg interval for IOP is present in all studies. Hypotonia (IOP under 5 mm Hg) is considered a therapeutic failure as it can lead to a decrease in visual acuity by choroidal folds and, more importantly, it may lead to phthisis bulbi.

As a response to this comment we added, in the “Discussion” section (lines 233-242) the following phrases: “Some studies define success as a decrease in IOP of more than 20% from the baseline value, while others, of more than 30%. In this study, it was opted for the latter, as we considered that a more important decrease in IOP is indicated in the case of these patients. This is because, after penetrating keratoplasty, it is difficult to assess the severity and progression of glaucoma by the use of conventional methods (clinical examination of the optic nerve and of the nerve fiber layers, automated perimetry, Optical Coherence Tomography) due to the partial or total opacification of the corneal graft and to a low visual acuity. In this situations, IOP is one of the few objective parameters that can be used for follow-up, although even IOP measurement is subject to many sources of error [14].”

  1. Comment: “Mention should be made regarding the type of glaucoma and severity, preferably using visual field results and or other widely used glaucoma staging systems to better comprehend the cohort considered.”

Response

The type of glaucoma, more precisely, the pathogenic mechanism, is mentioned for each patient in Table 1. Most cases had mixt etiologies and a comparative analysis regarding the glaucoma type could not be obtained, as it was difficult to divide the patients into distinct groups.

Regarding the severity, post penetrating keratoplasty glaucoma is a particular kind of glaucoma. This group of patients could not be evaluated using the methods used in most of the other glaucoma studies, for several reasons:

- The cornea was opaque in 30.7% of our patients, so perimetry and fundus examination was impossible

- Visual acuity was poor (important astigmatism, concomitant macular oedema, corneal oedema), which also impeded perimetry

As a result, IOP remained the only quantifiable parameter that could be determined with satisfactory precision in all cases. Whenever possible, patients were evaluated by automated static perimetry and optic nerve evaluation (clinical and by OCT).

In certain situations, we do not dispose of all the criteria necessary for making the diagnosis of glaucoma (such as optic nerve and visual field evaluation) and therapy is guided by the IOP level alone. Ocular hypertension after penetrating keratoplasty should be promptly treated, as it may lead to graft failure, even if it may not necessarily be associated with optic nerve atrophy. In cases where we could not evaluate the optic nerve or the visual field, we treated all IOP rises of more than 20% from baseline.

As our patients were diverse and complex we considered that presenting this type of data would not bring additional information of clinical significance, while statistical analysis would be influenced by the important number of patients with incomplete data sets. 

  1. Comment: “Details regarding prospective future studies in this field should be included in the Discussion section considering the small group of patients followed for a short time period and the retrospective nature of the study.”

Response

We added in the “Discussion” section (lines 316-319 ): “Moreover, future prospective studies of MP-TSCPC in post PK patients, with the inclusion of a greater number of patients, with standardized criteria for patient evaluation and with longer follow-up periods are needed in order to provide results with superior statistic relevance.”  

We thank you.

Yours respectfully,

the authors

Reviewer 2 Report

This is an article entitled “Micropulse transscleral cyclophotocoagulation for glaucoma after penetrating keratoplasty (diagnostics-1685936)” which evaluates the surgical outcome of micropulse transscleral cyclophotocoagulation in patients presenting with glaucoma after penetrating keratoplasty.

English needs revision

Abstract

  • Ok.

Introduction

  • Please also tell about the glaucoma drainage devices used in post-keratoplasty glaucoma such as Ex-PRESS, Ahmed Glaucoma Valve etc.

Materials&Methods

  • Did you measure the corneal endothelial cell count?
  • How long was the post-operative treatment regimen? Please admit.
  • Were all sutures removed prior to MP-TSCPC? Please admit.

Results

  • Please check and correctè “The gender distribution was: 119 15 men (57.7%) 11 women (42.31%).” The sum of the percentages does not make 100.
  • Are you sure that bullous keratopathy patients were not actually Fuchs corneal dystrophy? How did you decide? By checking the other eye or how?
  • Please give the standart deviations of all data.

Discussion

  • Please discuss the glaucoma drainage devices in post-keratoplasty glaucoma such as Ex-PRESS, Ahmed valve etc.
  • The absence of corneal endothelial cell count is also a limitation of the study. Please also admit.

References

  • Ok.

Tables

  • Ok.

Figures

  • Ok.

Author Response

Answer to Reviewer

We would like to thank the reviewer for taking the time to read and asses our manuscript. We believe that the suggestions have helped us improve our study. We have addressed all the concerns that were raised. Modifications are written in red in the new version of the manuscript and we used the Track Changes function of the Microsoft Word. We hope that the revised version meets your requirements. 

Introduction:

  • Please also tell about the glaucoma drainage devices used in post-keratoplasty glaucoma such as Ex-PRESS, Ahmed Glaucoma Valve etc.

Response:

Glaucoma drainage devices are mentioned and briefly discussed in the “Introduction”. The reviewer made the same comment for the “Discussion” section, so we followed their advice and added a paragraph in the “Discussions” section dedicated to other surgical methods used in post penetrating keratoplasty patients, especially glaucoma drainage devices, alongside the corresponding references (lines 296-319):

“Other surgical methods used in the management of glaucoma in the setting of PK include trabeculectomy and glaucoma drainage devices. Trabeculectomy, especially when associated with antimetabolites, is effective at lowering the IOP. However, it increases the risk of graft failure, either by intraoperative injury to the corneal endothelium, or by antimetabolite toxicity, which may lead to corneal ulcers and opacification [6][7]. Glaucoma drainage devices, such as the Ahmed glaucoma valve, Molteno and Baerveldt, have long been used in patients with glaucoma after PK. They are effective at reducing IOP, with reported success rates of 71% at 5 years [23] and 70% at 10 years [24]. They appear to be more effective than trabeculectomy at lowering the IOP in post PK patients, but they are associated with greater loss of corneal endothelial cells [25]. Graft survival is endangered for several reasons. First, glaucoma drainage devices lead to alterations in the blood-aqueous barrier which lead to changes in the protein content of the aqueous humor [26]. Second, tube positioning in the anterior chamber may result in contact with the corneal endothelium, with subsequent damage and endothelial cell loss [27]. The percentage of clear corneal grafts varies between 26 and 55% [23][24]. The Ex-PRESS mini-shunt is also an option, proving non-inferior to trabeculectomy in terms of IOP lowering capacity [26]. In a study by Ates et al., the postoperative IOP was successfully reduced by 30% or more compared to the baseline values, and no negative effects on the corneal graft were noted [28]. Additional studies comparing these methods and MP-TSCPC would be useful in helping the surgeons decide which options are better suited for their particular patients. Moreover, future prospective studies of MP-TSCPC in post PK patients, with the inclusion of a greater number of patients, with standardized criteria for patient evaluation and with longer follow-up periods are needed in order to provide results with superior statistic relevance.”

Materials & Methods

  • Did you measure the corneal endothelial cell count?

Response: We did measure the number of endothelial cells by specular microscopy whenever possible. In 30.7% of cases the graft was opaque, impeding the measurement, but there were also relatively transparent corneas where cell counting was not possible, possibly due to a very low count. This is why we considered that a statistical analysis of the remaining cases where the measurement was possible was not relevant.

  • How long was the post-operative treatment regimen? Please admit.

Response: We added the duration of the post-operative treatment regimen in lines 89-90: “topical dexamethasone 0.1%, 4 times daily for 4 weeks and cyclopentolate 1%, 2 times daily, for 2 weeks”

  • Were all sutures removed prior to MP-TSCPC? Please admit.

We responded in lines 123-124: “Twenty-four (92.3%) of patients had had all corneal sutures removed before the MP-TSCPC procedure, while the remaining two still had the running suture.”

Results

  • Please check and correctè “The gender distribution was: 119 15 men (57.7%) 11 women (42.31%).” The sum of the percentages does not make 100.

Response: We corrected, the new percentages are 15 men (57.7%) 11 women (42.3%).”

  • Are you sure that bullous keratopathy patients were not actually Fuchs corneal dystrophy? How did you decide? By checking the other eye or how?

Response: We had two cases of Fuchs corneal dystrophy. One of them had never had cataract surgery, but presented with important corneal oedema. The fellow eye had typical changes at the slit-lamp examination and on specular microscopy. The other patient was monocular and had had cataract surgery for that eye, the other eye having been lost in a childhood accident. However, the medical records prior to the cataract surgery showed a diagnosis of Fuchs corneal dystrophy.

In the cases of bullous keratopathy, all eyes had undergone cataract surgery and the fellow eye presented normal corneal characteristics on specular microscopy.

  • Please give the standard deviations of all data.

The Division of Biostatistics at our university, who was in charge of the statistical analysis of the data, decided to present the results as medians and IQR and not as mean± SD, because they believed that it was more correct, given the fact that we had a small number of patients and who did not have a gaussian distribution. At the request of the reviewer, they also provided the results expressed as mean± SD (Table A).

Table A. Description of analyzed variables

Variable

Average ± SD

Median (IQR)

Min

Max

Age

61.88 ± 15.07

67 (54.75-71.25)

25

80

Follow-up period

14.54 ± 3.467

13.5 (12-15)

12

24

IOP – Baseline

29.69 ± 4.174

29 (26-32)

25

40

IOP – 1M PostOp

18.62 ± 4.834

19 (16-21.25)

8

30

IOP – 3M PostOp

17.65 ± 4.372

17 (15.75-19.25)

8

28

IOP – 6M PostOp

17.46 ± 5.708

16.5 (13.5-20.5)

9

28

IOP – 9M PostOp

17.15 ± 4.994

16 (13.75-18.25)

12

31

IOP – 12M PostOp

18.42 ± 4.876

18 (16-20.25)

3

30

IOP – Baseline/12M Difference

11.27 ± 5.668

10 (7.75-12.25)

3

29

No. medication – PreOp

2.62 ± 1.061

3 (2-3)

0

4

No. medication – 1M PostOp

1.85 ± 1.287

2 (0.75-3)

0

4

No. medication – 3M PostOp

1.38 ± 1.235

1.5 (0-2)

0

4

No. medication – 6M PostOp

1.46 ± 1.392

1.5 (0-3)

0

4

No. medication – 9M PostOp

1.54 ± 1.272

2 (0-3)

0

4

No. medication – 12M PostOp

1.54 ± 1.208

1 (0.75-3)

0

4

No. med. – Baseline/12M Difference

1.08 ± 0.977

1 (0-2)

0

3

Discussion

  • Please discuss the glaucoma drainage devices in post-keratoplasty glaucoma such as Ex-PRESS, Ahmed valve etc.

We responded in the first comment (regarding the introduction).

  • The absence of corneal endothelial cell count is also a limitation of the study. Please also admit.

Response: We admitted in the “Discussions” section (lines 325-327): “Another limitation of the study resides in the fact that it did not analyze the effect of the procedure on the number of corneal endothelial cells, but only the clinical appearance of the graft.”

We thank you.

Yours respectfully,

the authors